# Improving Safety through Non-Technical Skills in Chemical Plants: The Validity of a Questionnaire for the Self-Assessment of Workers

**DOI:** 10.3390/ijerph16060992

**Published:** 2019-03-19

**Authors:** Marco Giovanni Mariani, Michela Vignoli, Rita Chiesa, Francesco Saverio Violante, Dina Guglielmi

**Affiliations:** 1Department of Psychology, Alma Mater Studiorum—University of Bologna, 40100 Bologna BO, Italy; rita.chiesa@unibo.it; 2Department of Psychology and Cognitive Science, University of Trento, 38068 Rovereto TN, Italy; michela.vignoli@unitn.it; 3Department of Medical and Surgical Sciences, Alma Mater Studiorum—University of Bologna, 40100 Bologna BO, Italy; francesco.violante@unibo.it; 4Department of Education Studies, Alma Mater Studiorum—University of Bologna, 40100 Bologna BO, Italy; dina.guglielmi@unibo.it

**Keywords:** workplace safety, non-technical skills, psychometric properties

## Abstract

This research is aimed at developing a questionnaire for the self-assessment of non-technical skills (NTS) leading to safety in the chemical sector and at analysing the properties of its scales in terms of construct validity. The research involved 269 Italian employees from three chemical plants of an international company, who occupied low–medium levels in the organizational hierarchy. Results showed a good level of validity and reliability of the instrument and suggested that communication, situational awareness, decision-making, and fatigue/stress management are the four most important NTS for safety in the chemical sector.

## 1. Introduction

As reported by the International Labour Organization [1], the chemical sector shows one of the highest large potentials of adverse effects such as (a) health hazards (e.g., cancer); (b) physical hazards (e.g., flammability), (c) environmental hazards (e.g., widespread contamination and toxicity to aquatic life) and more importantly (d) fires, explosions, and other disasters.

On the one hand, workplace safety has been recently defined as “an attribute of work systems reflecting the (low) likelihood of physical harm—whether immediate or delayed—to persons, property, or the environment during the performance of work” [2] (p. 2). Safety is a complex construct, which comprises many different elements that need to be taken into account. During the last few decades, the individual’s contribution to safety has mostly been studied in a negative way; indeed, safety science has focused on human error as the primary cause of accidents and incidents [3].

Recently, Beus et al. [2] defined an integrated safety model, which summarized the existing knowledge on safety theories. This model highlighted how safety knowledge and skills are relevant at an individual level in order to increase safety-related work behaviours and consequently, reduce accidents. Thus, to reduce the number of incidents and injuries, organisations need to improve levels of safety not only by improving technical equipment but also by enhancing the safety skills of workers.

In general, a skill domain refers to the ability to do something well (what a worker can do); specifically, safety skills can be classified into two main categories: technical and non-technical ones. While the first category refers to skills related to classical job activities (e.g., how to handle hazardous dust), non-technical skills (NTS) have been defined as cognitive, social and personal skills, complementary to technical skills, which contribute to safe and efficient performance [4]. One example of NTS in the chemical sector is communication skills, which allow for effective and appropriate exchange between the members of the work team, especially during shift changes, in order to avoid potential safety risks.

The published studies [5] demonstrated how relevant NTS are in order to create safer workplaces and prevent injuries. Despite a great amount of literature on non-technical skills, there are no instruments available to detect and study non-technical skills in the chemical sector, which is a relevant sector in terms of safety outcomes.

In order to address this important gap, this paper aims to define the main NTS for chemical sector workers (in particular in chemical plants) and to develop and validate a self-assessment instrument to measure them. Therefore, to accomplish this aim, the rationale of the procedure used was the identification of the most important NTS of the chemical sector, the production and selection of items and the evaluation of the psychometric properties of the questionnaire.

## 2. The Non-Technical Skills

### 2.1. The Domain of NTS

Yule and colleagues [6] refer to NTS as unspoken skills, not explicitly considered in work organization and in a formal socio-technical system, that can support the work performance adjustment and are crucial in incident management. Accordingly, NTS may be applied in many kinds of work environments because they are less domain-specific and more generalizable than technical ones [4]. For example, based on the non-technical skills for a given job, the Crew Resource Management training and assessment programmes approach was first introduced to the aviation sector in the 1980s and, subsequently, has been applied to acute medicine, offshore oil and gas operations too [4]. Obviously, the contents and the exercises of the training program, as well as the indicators for the assessment, were adapted to the specific domains but the general framework was substantially the same.

Flin and colleagues [4] noted that NTS contribute to the activation of safe and efficient task performance and developed a widely adopted taxonomy, comprising seven different non-technical skills described below: situational awareness, decision-making, communication, team work, leadership, stress management, and coping with fatigue (Table 1).

However, not all authors agree on the same model and approach; for instance, Thomas [3] claims that there is no definitive list of all the non-technical skills that can ensure safe operations and that non-technical skills depend on the industry context and the type of tasks that workers must carry out.

### 2.2. The Assessment of NTS

The assessment of NTS has become more and more relevant due to a growing number of professions that have begun to identify and train non-technical skills for safety-critical positions [4]. As reported by Flin et al. [4], the assessment of NTS is useful for many purposes as (a) giving feedback to trainees on their skill development; (b) testing the skills in an intervention program, analysing the needs of training; (c) studying whether an intervention has been effective (i.e., skills have been transferred to the job).

For some occupational sectors, there are already many instruments and tools to assess NTS. For example, a recent critical review found that 23 instruments aimed to assess NTS for individuals and teams working in the healthcare sector. Results of this review also highlighted how training and assessment of NTS were related to an improvement in safety and performance [11]. Although some instruments to assess NTS already exist in other sectors (e.g., Flin and Maran [12]), non-technical skills need to be slightly adapted to the work context in order to operationalize the construct. For example, communication between the members of a team is important in both the chemical and medical sectors. Nonetheless, the mere adaptation of the items from one context to another could lead to a potential lack of information due to the items not being focused on important elements for the safety in the new analysed context.

Consequently, more studies should focus on NTS’ assessment in high-risk sectors like the chemical industry.

## 3. Aims of Research

Despite the taxonomy developed by Flin and colleagues [4], many studies have investigated the basic domains of NTS, but so far, no definitive list of all the NTS has been available, as reported by Thomas [3], who supposed that NTS depended on the context and nature of the work performed [3]. Similarly, Flinn and Maran [13] conclude that NTS must be carefully specified for a given profession and set task. However, the general taxonomies of NTS are similar across professions [12].

Thus, the first aim of this study is the development of an instrument able to assess different types of NTS that are important for safety in chemical plants. The study focused on creating a brief and practical questionnaire (NTSC-Q) that measures this construct in a reliable, valid and efficient manner. The second aim is the validation of NTSC-Q. According to the American Psychological Association [13], psychometric properties were investigated in terms of construct validity and external validity. Some important aspects of internal validity will be analysed, such as dimensionality and reliability, while external validity will be tested through criterion-related validity. Given that internal validity focuses primarily on the internal relationships of the test, the first step will be to analyse whether the theoretical dimensionality of the scale is supported.

Therefore, our first hypothesis regards factorial structure and dimensionality:

**Hypothesis** **(H1):***The multidimensional model used to develop the scale fits the collected data better than the alternative models*.

In line with the first hypothesis, the associations between each item and the dimension it intends to measure, the association between the various dimensions and, finally, the reliability of each dimension will be verified.

Furthermore, we wanted to study the external validity and specifically, the criterion-related validity, that is the degree to which scores from our test correlate in expected ways with a network of measures that have previously been validated [14]. Criterion-related validity can be composed of concurrent validity and predictive validity.

With this aim of criterion-related validity, we referred to the model of Griffin and Neal [15] that supposes that knowledge/skills are independent variables of safety performance. Obviously, NTS can be considered an important element of this area that Campbell et al. [16] defined as the “can do” domain. However, human performance is affected not only by abilities (“can do”) but by motivational factors (“will do”) too. Thus, Griffin and Neal [15] affirmed that performance is also influenced by a motivational area that concerns why an employee should perform a task and the individual willingness to exert effort to adopt a behaviour (what a worker wants to do). Both motivation and skills are necessary to achieve high levels of safety performance. For this reason, we use safety motivation as a concurrent variable in order to evaluate the NTS construct validity.

Moreover, Griffin and Neal [15] supposed that safe performance encompasses compliance and participation. The first is related to the task (compliance) and the second to the context (participation).

On the basis of the model of Griffin and Neal [15], we hypothesized that:

**Hypothesis** **(H2):**With respect to concurrent validity, safety is positively correlated to NTS.

**Hypothesis** **(H3):**With respect to predictive validity, NTS positively affects the safety performance.

## 4. Methods

### 4.1. Participants

Three chemical plants of an international chemical company were involved in the study. The number of participants was defined on the basis of the statistical analysis required and a not random sample was adopted. All employees at a low–medium level in the organizational hierarchies were involved.

A total of 269 employees (87% of the total amount) completed the questionnaire on NTS and their responses were used to test the internal validity.

As far as their involvement in work safety is concerned, 9% of participants declared they held a role of “supervisor in the safety domain,” 4% were “delegates of the employees in the safety domain,” and the remaining employees were simple workers with no specific role in safety. Furthermore, 14% of participants declared they were “responsible in case of emergency.”

One hundred eighty-three participants had the time and opportunity to fill in a second section of the questionnaire with more concurrent and predictive variables for studying external validity. They had roughly the same characteristics as the whole sample.

The pyramid of Heinrich [17,18] for the last three years has shown the following data: 11 first aid incidents, seven lost time accidents and zero severe (fatal) injuries in the analysed organisations.

### 4.2. Procedure and Measure

The procedure of this study is divisible in two phases; the first allowed the construction of the questionnaire and the second is a validation of the same. As Figure 1 shows, first the NTSs were identified, followed by the production of the items; then came the collection of the data using the questionnaire and, finally, analysing the results to study the validity.

The data were collected by a structured anonymous paper/pencil questionnaire. The study assured respondents’ anonymity and confidentiality. The questionnaire included a statement regarding personal data treatment, in accordance with Italian privacy law. All questionnaires were collected in a group of 8–12 participants, before convening sessions on safety. As the collected questionnaire was anonymous and the research investigated psychosocial variables, not adopting a medical perspective, ethical approval was not sought.

In respect to the measures, apart from the NTSC-Q used to measure NTS in the chemical sector, of which development and validity will be presented in the results section, in this study we measured:

#### 4.2.1. Safety Motivation

For concurrent validity analysis, we adopted the questionnaire of Mariani et al. [14] to measure the safety motivation as an individual willingness to exert effort to adopt safety behaviour and the valence associated with that behaviour [19]. Two scales were filled in by participants: the first regarded extrinsic motivation, which refers to being motivated by obtaining a reward or by avoiding punishment or injuries (four items, e.g., “I behave in a safe way in order to avoid being criticized by others”); the second regarded intrinsic motivation, which means that employees safely perform an activity because they perceive it as interesting, satisfying or pleasurable (four items, e.g., “I behave in a safe way because it makes me feel satisfied”). A five-point Likert scale was used to record the participant’s opinion.

#### 4.2.2. Safety Performance

For predictive validity analysis, Toderi et al.’s [20] Italian version of the Griffin and Neal model’s scale was adopted. It measures the components of the safety performance—compliance, relatedness to the task, and participation—concerning the context. Compliance performance refers to behaviours that maintain the safety of the work environment (e.g., compliance with safety protocols/procedures). Participation performance refers to behaviours, mostly voluntary, that indirectly contribute to the safety of the work environment. Both of them are important in developing attention and social value to safety within the organizational context (e.g., participation in safety initiatives on a voluntary basis). Each measure consisted of four items that used a five-point Likert scale to record the participant’s opinion.

### 4.3. Statistical Analysis

#### 4.3.1. Internal Validity

The dimensionality was analysed by confirmatory factor analyses (CFA), comparing three alternative measurement models: the hypothesized four-factor model with covariance among factors, a one-factor model and a four-factor model without covariance among factors. CFAs were conducted using IBM SPSS Amos 22.

The estimation procedure was performed by the maximum likelihood method. In structural equation models, there are several Fit Indexes that reflect how the model is in line with the data; Hair et al. [21] recommend the use of at least one fitness index from each category of model fit. Thus, we adopted RMSEA (Root Mean Square of Error Approximation) for testing absolute fit, CFI (Comparative Fit Index) for the incremental fit, and, finally, Chisq/df (the ratio of the model χ² and the degrees of freedom) to analyse the parsimonious fit.

Schweizer [22] summarizes the acceptable levels of fit indexes: normed χ² is expected to stay below 3, RMSEA below 0.08, and the CFI value should be in the range of 0.90 to 1.00.

On the one hand, Akaike Information Criterion (AIC) [23] and the Bayesian Information Criterion (BIC) [24] fit indexes were used to compare the non-nested models of CFA (a four-factor model with covariance among factors against a one-factor model). The AIC permits us to select a model on a measure of similarity between a correct model and competing models. The BIC is to choose the model with the highest probability of being the correct model. AIC and BIC give penalties to models with more parameters and, therefore, help to protect against unnecessary model complexity. The smaller the AIC and BIC measures are, the better the fit. On the other hand, to compare the nested models (a four-factor model with covariance among factors against a four-factor model without covariance among factors) the χ^2^- difference test was used. A non-significant value of this statistic means that the overall fit of two models is comparable [25]. In this case, the model with the highest degree of freedom is the better choice, as it is the most parsimonious model [26].

The reliability analysis was conducted to evaluate whether the NTSC-Q items were measured in keeping with the study’s intention. We adopted Cronbach’s alpha. As suggested by Hair et al. [21], when an index is greater than 0.70, sufficient construct reliability is obtained. The Average Variance Extracted (AVE) index was used to verify the Validity of every construct. AVE is an indicator of convergent validity; indeed, it shows the amount of variance explained by the construct in relation to the variance due to error: AVE values greater than 0.50 indicate good convergent validity [21].

Bentler and Chou [27] suggested that the size of a fair sample is more than 15 times the number of observed variables for a CFA. Our model has 16 observed variables, so it would need a minimum sample of 240 subjects.

#### 4.3.2. External Validity

In order to study the external validity of the NTSC-Q, we evaluated its relationship with an antecedent of NTS (safety motivation) and one of their outcomes (Safety Performance). The structural equation model approach was adopted to test global relationships among Motivation NTS and Performance. Moreover, the single correlation indexes between the variables were computed for a deeper analysis. A minimum sample of 120 participants is fair for this analysis on the basis of Bentler and Chou [27].

## 5. Results

### 5.1. Development of NTSC-Q Scale

Following the approach of Flin et al. [4], the identification of NTS was based on various forms of task analysis. Firstly, accidents and near-misses were studied to find the behaviours that were involved in these events. The second source of information was the procedures and rules for safety. These were studied so as to note the most relevant behaviours for safety. A third source of information was based on interviews. Twelve key informant interviews were conducted with employees of three chemical plants with different roles in safety. Two occupational doctors, three HSE (Health Safety and Environment) managers, three production/line managers, and four workers’ representatives for health and safety at work were interviewed to collect data not only on relevant safety behaviour, but on errors and organizational misbehaviour, as acts in the workplace that are done intentionally and constitute a violation of rules, too. All interviews ended with a visit to the plant to explain, with practical examples, what had been described. Following the suggestions of Morse [28], 12 interviews were considered sufficient as the researchers agreed that data saturation had been reached, which meant that nothing new would be added to our understanding of the research topic by collecting further data.

Then, four psychologists, experts on safety and chosen because they had already worked on NTS in chemical plants, categorized the behaviours that had been collected from the three different sources, with the NTS taxonomies of Flin et al. [6]. Each behaviour was associated with a NTS of the model, therefore NTS taxonomies were considered complete and exhaustive and it was not necessary to identify or add new categories.

Table 2 presents the results of this task.

Considering that we wanted to develop a short and efficient questionnaire, the most important NTS taxonomies for the safety of workers in chemical plants were identified as: Communication, Situational awareness, Fatigue Management, and Decision-Making. Leadership and Teamwork were not as significant for safety in chemical plants, and had the lowest positions.

The four taxonomies can be described as follows. Indicators of Situational awareness refer to knowledge of what is going on around an employee. It is the first of the cognitive skills that must be considered. It consists of constant monitoring in the workplace, observing what is going on and detecting any changes in the environment [4].

Decision-making refers to the process of reaching a judgement or choosing an option. It includes the skills associated with the continuous cycle of monitoring and re-evaluating the task environment in order to take appropriate action.

Further indicators have been developed to measure Communication skills as a basic element for good teamwork and workplace efficiency and safety. They are based on the exchange of information, feedback or response, ideas and feelings [4].

Finally, behavioural indicators concerning the skills to face fatigue have been identified. Fatigue is generally considered a decline in mental and/or physical performance on the basis of prolonged exertion, sleep loss and/or disruption of the internal clock. Research reveals that when we are sleep-deprived and/or fatigued, performance is affected and errors are more likely to occur [29].

Later, 8–10 items were produced for each taxonomy and then we chose the best for each NTS. Experts followed the procedure presented by Grimm and Widaman [13] with respect to content validity, that is, the degree to which items‏ of‏ an instrument sufficiently represent the content domain. The best four items of each NTS were selected: every item followed the Grimm and Widaman [13] criteria, which reflect that more than half of the judges indicate the item as “essential.”

Table 3 shows the best four items of each NTS. The instructions for completing the questionnaire and the adopted scales for collecting the answers are reported in notes.

### 5.2. Internal Validity: Dimensionality and Reliability

Confirmatory Factor Analysis was conducted and the results are reported in Table 4. The one-factor solution provided an inadequate fit for the data. A similar result was found with the four-factor model without the covariances among the latent dimensions. By contrast, the four-factor model including the covariances among the latent dimensions showed an acceptable fit. The RMSEA index was 0.07 and CFI 0.96. Moreover, the value of Chisq/df was 2.2. The model got the lowest level in AIC and BIC as a further check of a good fit with respect to the remaining models. However, the χ^2^ difference between a four-factor model without vs. with the covariances between the factors confirmed a significant improvement of fit (Δχ^2^ = 455.57; Δd.f. = 6; *p* < 0.001), making the third model in Table 4 preferable. Therefore, the hypothesized structure of H1 was confirmed.

In order to verify that the items designed to measure each dimension do so in the expected ways, factor loadings were analysed (Table 5). All of them were greater than 0.60 (0.66–0.98) and the AVE values were greater than 0.50 (0.58–0.65). All this confirmed the convergent validity, as suggested by Hair et al. [21].

Moreover, regarding the association between the dimensions, the results showed that the correlation indexes between factors had a range between 0.59 and 0.81. Only one correlation index showed a common variance higher than 50%. It concerned the covariance of Decision-making and Fatigue management factors. Thus, we tested a further model in which the item of the last factors depended on only a latent variable (model 4 in Table 5). The model with only three factors showed an inadequate fit on the basis of every analysed index. Consequently, we accepted model 3, which hypothesized a structure with four correlated factors.

Then, the reliability of each dimension was tested. Cronbach’s alpha showed higher values than 0.70 (0.85–0.91), proving good reliability for every dimension.

Finally, the scores of the four scales were calculated. Table 6 shows descriptive statistics of the NTSs for every plant. 

A MANOVA with four dependent variables (the scores of NTS scales) and one fixed factor that considered the three plants was carried out. The multivariate analysis showed that there were no significant differences between the plants in the level of NTSs (Pillai’s Trace = 0.019, F = 0.71, df = (8584), *p* = 0.682).

### 5.3. External Validity

To verify external validity, we tested a model that considered, with respect to NTS, a concurrent variable, Safety Motivation, and a dependent variable, Safety Performance. Table 7 shows the correlation matrix of the variables that were adopted for the analysis of external validity.

The model of external validity presented satisfying fit indexes: Chi square = 35.82; DF = 17; Cmin/DF = 2.11; RMSEA = 0.07 and CFI = 0.96 (Figure 2). The standardized covariance index between NTS and Motivation was 0.60 and that confirmed the concurrent validity and, consequently, H2. The standardized regression index between NTS and Performance was 0.40, consistent with the predictive validity and H3.

## 6. Discussion

NTS are an important factor for safety and accident prevention in the workplace; our aim was to contribute to developing and validating a self-assessment instrument to measure them in this sector.

The results showed that the behaviours that had been collected, on the basis of the three different sources, were categorized in the NTS taxonomies of Flin et al. [6]. Each behaviour was associated with a NTS of the model; therefore, we may consider the NTS taxonomies of Flin et al. [6] complete and exhaustive for this research.

Moreover, the results presented Communication, Situational awareness, Decision-making, and Fatigue/stress management as the four most important NTS in all three chemical plants involved in the study. A good exchange of information is the first step for a safe system for workers in the chemical sector, followed by Situational awareness, which permits active monitoring of the workplace. The important role of Decision-making in a chemical plant is recognisable as every worker’s task deals with decisions that have an impact on safety. Finally, the findings show that Fatigue management is another important NTS in chemical plants, due to the fact that employees have to work night hours because of their work shifts.

Notably, leadership did not seem so important; however, it must be taken into consideration that the participants to this study were workers without subordinates, and team leaders were not involved.

The results of the validity analysis confirmed the multidimensional structure: in line with our first hypothesis, the four-factor model with the covariances among the latent dimension reported the best fit with the data. In the scientific literature, few researchers studied the psychometric proprieties of NTS scales and only some adopted confirmatory factor analysis. However, studying surgeons’ non-technical skills, Yule et al. [6] validated an assessment tool that analyses four categories of NTS: Situational awareness, Decision-making, Communication and Leadership. Our CFA was better than theirs with respect to both fit indexes, CFI and RMSEA.

Furthermore, all NTS measured have good reliability, factor loadings and the items measure each dimension in the expected ways. All that is in line with the results that Yule et al. [6] found in their research. Moreover, we conducted a MANOVA analysis that showed no differences in the NTS levels between the three plants. This can be explained by the fact that all three plants were part of the same international chemical company and adopted the same management approach for safety and the same personnel selection procedures and training program for all employees.

Concerning the external validity, we utilized the Griffin and Neal theoretical framework [15]. According to our hypothesized model, motivation, understood as an individual willingness to exert effort to adopt safety behaviour and the valence associated with that behaviour [19], was expected to be associated with NTS (concurrent validity), which in turn are associated with safety performance (predictive validity). Our results confirmed both these relationships. Specifically, regarding predictive validity, NTS are linked to two components of safety performance, compliance and participation. We can compare our evidence with the results of the Griffin and Neal model, which adopts the knowledge concept instead of NTS. However, the alphas of our measures are in line with the evidence from the literature [17,18]; moreover, the path coefficients between NTS and motivation are comparable with the results of Barbaranelli, Petitta, and Probst [30], who adopted a double sample, one from Italy and one from the USA. In addition, the path coefficients we found are higher than Griffin and Neal’s [16]. The path coefficient of the relationship between NTS and performance is substantially in line with the results of Barbaranelli, Pettita and Probst [30], and Griffin and Neal [15].

In addition to confirming the validity of the instrument, these results highlighted how NTS are relevant in order to increase safety-related work behaviours at an individual level and, consequently, they suggest that NTS can reduce accidents, in line with the model regarding knowledge and skills proposed by Beus et al. [2]. So, overall, our findings highlight the main NTS for the chemical sector and present a method for assessing them, with high levels of validity.

However, there are some limitations to consider in this research. Firstly, the respondents belonged to a single multinational company; this could limit the generalizability of the results in other contexts.

Secondly, we focused on the main NTS that were confirmed by three sources of data. Everything gave rise to a parsimonious four-category model, but we think that further analysis is needed, especially if the employees’ roles were to change and new technologies were adopted at work, to consider NTS missing in this study.

Thirdly, the cross-sectional nature of our data prevents us from clarifying the causal relationship between the variables under investigation. Future studies should also investigate the hypotheses in a longitudinal way. Moreover, the current data were collected with self-reported measures, thus increasing the likelihood of a common method variance effect; therefore, future studies should investigate the NTS with objective or observed data. Furthermore, the research does not adopt an experimental design that can control any variable and study the effects of them on NTS. This is a correlational study with a qualitative/quantitative research design that presents these limitations but can achieve a high level of generalization with respect to the findings too.

Finally, this study involved one country, so there is a need to replicate this study in other cultural contexts. Although the means and standard deviation of the items do not imply a need to improve NTS in a certain context, their references could still be useful in order to compare the organizational data collected by different organizations.

Despite these limitations, our results confirmed that NTSC-Q can be an instrument with satisfactory psychometric properties useful to evaluate the NTS in the chemical sector. Furthermore, another strength of this study is that, instead of merely adapting existing scales of NTS in other sectors, we tried to develop a new one based on a qualitative investigation. This is also in line with what Flin and Maran suggested [12]: that it is inadvisable to use a non-technical skill set devised for one occupational context (e.g., aviation) in a different work setting (e.g., healthcare).

## 7. Conclusions

We believe that this questionnaire could be used in different ways within the safety and prevention fields, for instance, in order to evaluate the training needs concerning the NTS and to evaluate the integration of safety management plans. In particular, since NTS are important to enhance workers’ safety performance, they can be developed and implemented. In line with the Griffin and Neal framework [15], safety training represents a key factor in determining safety outcomes, specifically safety compliance and safety participation.

The NTSC-Q could be a useful tool for organising a project training program (see, for example, Mariani et al. [31], who utilised the NTS evaluation in this way). This tool still does not have a normative sample that allows its diagnostic use, but it allows for NTS identification within an organisation to be developed through the training program. Indeed, NTSC-Q is not suitable for clinical assessment of a single person but for analysis at an organizational level, from the perspective of injury prevention and safety.

Therefore, the NTSC-Q enables the evaluation of four NTS that have been identified as relevant to the chemical sector and could be utilised for training planning, as in other sectors that use specific tools (see, for instance, Yule et al. [6] for the health sector). In the near future, the NTSC-Q could be used to compare different organisations relative to safety performance and could be integrated with further clinical/diagnostic tools to assess the NTS at an individual level, too.

## Figures and Tables

**Figure 1 ijerph-16-00992-f001:**
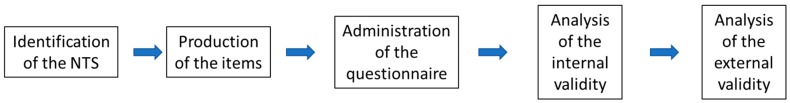
The procedure of the research.

**Figure 2 ijerph-16-00992-f002:**
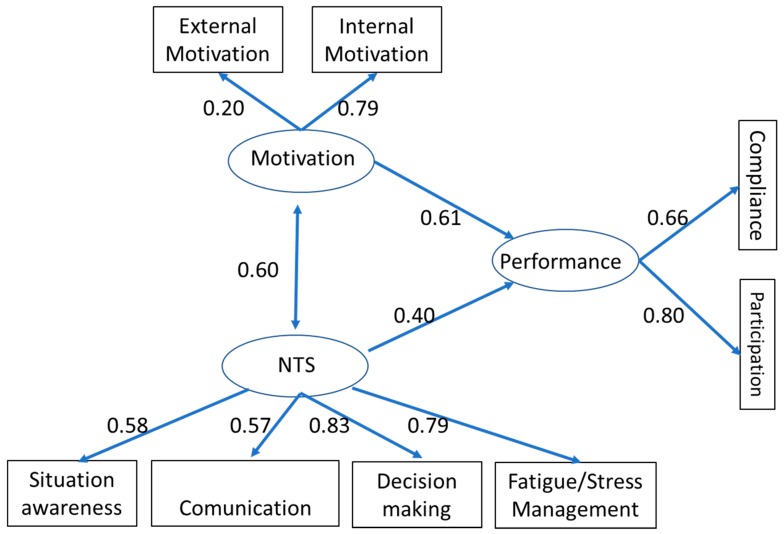
The concurrent and predictive validity (*N* = 183).

**Table 1 ijerph-16-00992-t001:** Flin and colleagues’ [4] NTS taxonomy.

*Situational awareness* Situational awareness is defined as knowing what is going on around an employee and mainly consists of constant monitoring of the workplace by observing what is going on and detecting potential changes in the environment [4]. One of the most recognised models of situational awareness was developed by Endsley [7] and is composed of three levels: (1) information collection (perception); (2) interpretation of the collected information (comprehension); (3) anticipation of future states (prediction).
*Decision-making* This can be described as the process of reaching a judgement or choosing an option. Decision-making is the NTS most directly related to behaviours. Usually, in operational work settings, there is a continuous cycle of monitoring and evaluating the task environment, then the employees have to take appropriate action. Specifically, Flin and colleagues [4] identified a four-step process composed of: (1) evaluation of the situation and definition of the problem; (2) creation and consideration of one or more response options; (3) selection and implementation of one option; (4) results’ revision.
*Communication* Communication is a basic element of a good team work and it is vital to workplace’s efficiency and safety. It refers mainly to four elements: (1) sending information in a clear and concise manner; (2) including context variables and the intent of the exchange of information; (3) receiving information, especially through listening; and (4) identifying and re-directing the communication barriers [4].
*Teamwork* Teamwork is a relevant NTS as working in teams is an important characteristic of modern organisations. In fact, nowadays, teams are more frequently composed of people with different expertise and capabilities. Four main elements for teamworking have been identified by Flin and colleagues [4]: (a) support from others; (b) conflict resolution; (c) exchange of information; (d) activity coordination.
*Leadership* Leadership has been defined as the skill of getting others to do (and want to do) something that the leader believes should be (must be) done in a safety perspective. It involves interpersonal influence, goal-setting, and communication [8].
*Stress management* Stress management refers to the ability to manage stress and stressful situations in the workplace. There is a growing number of studies that have investigated the role of stress in determining safety outcomes. For example, Li and colleagues [9] found that job demands (psychological and physical ones) and job resources (decision latitude, supervisor support, and co-worker support) may affect emotional exhaustion and safety compliance, and, thus, they may influence the occurrence of injuries and near-misses. This highlights the importance of stress management as a relevant NTS in order to prevent negative safety outcomes.
*Coping with fatigue* It is especially important in the chemical sector as many companies work 24 hours, seven days a week. Therefore, in this sector, workers need to be trained to better manage fatigue as this has been acknowledged as a significant safety concern [10]. Fatigue is generally considered as a decline in mental and/or physical performance based on prolonged exertion, sleep loss and/or disruption of the internal clock. Flin and colleagues [4] defined three main elements on the NTS concerning coping with fatigue: (1) identification of the antecedents of fatigue; (2) identification of the consequences of fatigue; (3) implementation of coping strategies.

**Table 2 ijerph-16-00992-t002:** Ranking of NTS for safety with respect to sources of information.

	Accidents and Near-Misses	Procedure and Rules for Safety	Interviews
1	Communication	Communication	Communication
2	Situational awareness	Decision-Making	Fatigue Management
3	Fatigue Management	Situational awareness	Situational awareness
4	Decision-Making	Fatigue Management	Decision-Making
5		Teamwork	Teamwork
			Leadership

**Table 3 ijerph-16-00992-t003:** Behavioural indicators measured by items of the scales.

Situational awareness	Indicate specific hazards involved in carrying out the tasks of the job
	Pay attention to details that may cause risks
	Monitor the situation to prevent possible hazards
	Predict probable future hazards
Decision-Making	Prioritize when decisions need to be made
	When required, make quick decisions
	Predict the effect of the decisions
	Identify and manage priorities
Communication	Communicate effectively with the supervisor
	Communicate effectively with colleagues
	Give information/feedback on your work
	Ask for information/feedback on your work
Fatigue	Recognise the states and causes of physical fatigue
	Recognise the states and causes of mental fatigue
	Implement strategies to cope with physical fatigue
	Identify sources of stress

Note: The items were introduced by the statement “Think of good health and safety practices at your workplace; how much do you feel able to contribute?”. Five-point scales were presented for the answers: Not able to at all, Slightly able, Moderately able, Very able and Extremely able.

**Table 4 ijerph-16-00992-t004:** Fit indexes of the models (*N* = 269).

Model	Df	χ2	χ^2^/df	RMSEA	CFI	AIC	BIC
1	104	807.81 ***	7.77	0.16	0.74	871.80	986.84
2	104	671.47 ***	6.46	0.14	0.79	735.47	850.50
3	98	215.90 ***	2.20	0.07	0.96	291.90	428.50
4	101	299.38 ***	3.00	0.08	0. 93	369.39	495.202

Note: 1 = One-factor model; 2 = Four-factor model without covariance among latent dimensions; 3 = Four-factor model with covariance among latent dimensions; 4 = Three-factor model where items of fatigue management and decision-making were affected by a single factor. *** *p* < 0.001.

**Table 5 ijerph-16-00992-t005:** Reliability and validity statistics for the four-factor model (*N* = 269).

Factor	Items	Mean	Standard Deviation	Loadings	AVE	Cronbach’s Alpha
Situational awareness	Indicate specific hazards involved in carrying out the tasks of the job	4.04	0.77	0.66	0.59	0.85
	Pay attention to details that may cause risks	4.10	0.80	0.82		
	Monitor the situation to prevent possible hazards	4.06	0.82	0.81		
	Predict probable future risks/hazards	3.80	0.88	0.77		
Communication	Communicate effectively with the supervisor	3.89	0.95	0.83	0.72	0.91
	Communicate effectively with your colleagues	3.99	0.89	0.83		
	Give information/feedback on your work	4.02	0.88	0.86		
	Ask for information/feedback on your work	3.86	0.89	0.88		
Decision Making	Prioritize when decisions need to be made	3.70	0.98	0.77	0.58	0.85
	When required, make quick decisions	3.89	0.96	0.79		
	Forecast the effects of decisions	3.71	0.89	0.78		
	Identify and manage priorities	3.83	0.89	0.71		
Fatigue management	Recognize the states and causes of physical fatigue	3.80	0.92	0.73	0.65	0.88
	Recognize the states and causes of mental fatigue	3.75	0.97	0.78		
	Implement strategies to cope with physical fatigue	3.70	0.91	0.84		
	Identify sources of stress	3.72	0.97	0.87		

**Table 6 ijerph-16-00992-t006:** Descriptive statistics of the scales.

	Total Sample (*N* = 269)		Plant 1 (*N* = 125)		Plant 2 (*N* = 121)		Plant 3 (*N* = 123)	
	M	D.S.	M	D.S.	M	D.S.	M	D.S.
1. Situational awareness	3.99	0.68	3.91	0.68	4.01	0.70	4.04	0.66
2. Communication	3.95	0.80	3.88	0.81	3.97	0.79	4.00	0.79
3. Decision-Making	3.75	0.79	3.67	0.74	3.84	0.76	3.74	0.88
4. Fatigue management	3.75	0.80	3.68	0.76	3.76	0.85	3.81	0.80

**Table 7 ijerph-16-00992-t007:** Correlations (Pearson’s *r*) of the scales (*N* = 183).

	1	2	3	4	5	6	7	8
1. Situational awareness	(0.85)							
2. Communication	0.46	(0.91)						
3. Decision-Making	0.54	0.51	(0.85)					
4. Fatigue management	0.48	0.42	0.63	(0.88)				
5. Extrinsic motivation	0.05	0.12	0.12	0.01	(0.73)			
6. Intrinsic motivation	0.18	0.29	0.42	0.36	0.16	(0.84)		
7. Safety compliance	0.37	0.34	0.47	0.51	0.17	0.39	(0.81)	
8. Safety participation	0.22	0.39	0.50	0.44	0.07	0.58	0.56	(0.85)

*Note*. Cronbach’s alpha in brackets. The scores of all variables were computed as the average among items.

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
