# Peer review of "Improving Safety through Non-Technical Skills in Chemical Plants: The Validity of a Questionnaire for the Self-Assessment of Workers"

_ijerph, 2019, doi:10.3390/ijerph16060992_

Round 1
Reviewer 1 Report
- Regarding the development of NTSC-Q scale, did the method include a list of NTS to be included or not included? It seems that the three sources of NTS are not related to the NTS identified in other fields. If so, can it be possible that some NTS were not identified just because the participants did not know they exist? Furthermore, NTS taxonomy is new or selected from the literature referenced? Or this research is only looking up for Table 1 NTS?
This concern is related to the Discussion 326 and following. The research is strong in the meaningfulness of the latent variables related to the four NTS but I differ that the research demonstrates they are the most important, unless other NTS are considered. Moreover, is there any other NTS missing?
This research cannot be definitive as far as the method could be missing any other NTS. Please explain the weaknesses of the research related with my concerns in Discussion section.
- Please also report if there are significant differences between the three plants.
- Minor issues
In line 77 there is an indentation not needed.
In citing not use "and collegues" but "et al."
In line 365 the meaning in the first sentence is confusing, is not better to say: "However, there are some limitations in this research to consider"
Author Response
Comments and Suggestions for Authors
- Regarding the development of NTSC-Q scale, did the method include a list of NTS to be included or not included? It seems that the three sources of NTS are not related to the NTS identified in other fields. If so, can it be possible that some NTS were not identified just because the participants did not know they exist? Furthermore, NTS taxonomy is new or selected from the literature referenced? Or this research is only looking up for Table 1 NTS?
This concern is related to the Discussion 326 and following. The research is strong in the meaningfulness of the latent variables related to the four NTS but I differ that the research demonstrates they are the most important, unless other NTS are considered. Moreover, is there any other NTS missing?
This research cannot be definitive as far as the method could be missing any other NTS. Please explain the weaknesses of the research related with my concerns in Discussion section.
Answer:
Dear Referee, thank you for your suggestions.
We agree that the text doesn’t sufficiently explain this aspect of our research. We adopted the model of. Flin and colleagues [4], that is presented in table 1, to categorize the behaviours of the three different sources. Every behaviour was associated with a NTS, therefore we assume that the model of Flin and colleagues [4] was complete and it was not necessary to identify/add new taxonomies. The three sources were fairly in agreement on identifying the major NTSs, so we considered this a consistent result. However different studies have shown the NTS model with four categories (i.e. Yule and colleagues, [6[). We have tried improving the text by explaining this concept (see line 259 and 323) and we have also followed this up in the discussion section.
- Please also report if there are significant differences between the three plants.
There are no significant differences between the three plants. A MANOVA with four dependent variables (the scores of NTS scales) and one fixed factor, that considered the three plants, was carried out. The multivariate result was not significant for the plant factor, Pillai’s Trace = .019, F = 0.71, df = (8,584), p = .682, indicating no difference in the levels of NTSs between the employees of the three plants. We have added this integration to line 327.
- Minor issues
In line 77 there is an indentation not needed.
In citing not use "and collegues" but "et al."
In line 365 the meaning in the first sentence is confusing, is not better to say: "However, there are some limitations in this research to consider"
Thank you, we have also improved the text with these last minor suggestions and English has been reviewed.
Best Regards,
The authors
Reviewer 2 Report
Thank you for the detailed response. I have no further comments or questions.
Author Response
Thank you, for your feedback.
We have improved the English language.
Best Regards,
The authors
Reviewer 3 Report
Dear authors:
I have been reviewed carefully. It has been a pleasure to review your paper about the Validity of a Questionnaire for the Self-assessment of Workers but I have observed a lot of methodology errors and poor results.
1. The introduction is poor, needs to present a better rationale for the study and the methodology employed.
2.Overall the design was poor. Research question is not well designed and methods carried out with insufficient detail and information to replicate. Also, there does not appear to be any justification for the method used for calculation of simple size. The sample is very small to do the statistical analysis. The criteria of inclusion and exclusion were not included in the text. You did not include a flow diagram of the process. The statistical analysis is very simple, I would been liked to see the matrix of correlation or the initial factor matrix .
3. The discussion section is too simple.For example: How does the NTS compare to other measures in Italain. What differences are there? Why should the clinician choose the one before the other?. I can not see the population that you focus your research.
4.Results, Discussion and Conclusions sections are not informative. I dont believe this study adds a great deal of novel and new information.
I regret that the disposition is not favorable, but would like to thank you for your support.
We wish you all the best.
Author Response
Dear authors:
I have been reviewed carefully. It has been a pleasure to review your paper about the Validity of a Questionnaire for the Self-assessment of Workers but I have observed a lot of methodology errors and poor results.
Dear reviewer,
we carefully read your comments that helped the improvement of the final manuscript.
1. The introduction is poor, needs to present a better rationale for the study and the methodology employed.
We understand the request, but our main aim is to try to fill the gap related to there not being instruments available to detect and study non-technical skills in the chemical sector and we feel that we have satisfied this point. However, we have tried to improve the presentation of the rationale, in the introduction and in the method sections.
2.Overall the design was poor. Research question is not well designed and methods carried out with insufficient detail and information to replicate. Also, there does not appear to be any justification for the method used for calculation of simple size. The sample is very small to do the statistical analysis.
We have added further details on the procedure/method and we have bolded the rationale for defining the sample size.
The criteria of inclusion and exclusion were not included in the text. You did not include a flow diagram of the process. The statistical analysis is very simple, I would been liked to see the matrix of correlation or the initial factor matrix .
We think that the data matrix of the correlation between all items is too big for the standard of this publication, however you can read it the attached document.
3. The discussion section is too simple.For example: How does the NTS compare to other measures in Italain. What differences are there? Why should the clinician choose the one before the other?. I can not see the population that you focus your research.
We are sorry but there are no questionnaires on NTS in the chemical sector and therefore we are unable to compare the results. However, we have presented some suggestions on the basis of different studies on NTS, applied in different sectors.
4.Results, Discussion and Conclusions sections are not informative. I dont believe this study adds a great deal of novel and new information.
I regret that the disposition is not favorable, but would like to thank you for your support.
We wish you all the best.
Best Regards,
The authors

Round 2
Reviewer 3 Report
My comments to this manuscript are the same as before the revision. The problems remain the same.
I would like to thank the authors for their work, however I did not feel the authors made any significant improvements with regards to the main issues I raised in the first review. In its present state the paper provides no clear evidence that the authors propose a experiment must have been conducted rigorously. In addition, the clinical relevance is not high and need to be given rationale. Discussion and conclusion need to focus on the findings and its significance.
Author Response
We thank the reviewer for the last feedback. We have improved the paper following these lines:
1) A short integration of the presentation of the findings in the discussion
2) More details on limitations that focuses that we have not adopted an experimental design
3) A final specification that shows that NTSC-Q has not a clinical perspective for assessing the individual NTS but it is useful to study the NTS at organizational level, for training program.
So we think that the paper has been improved as much as possible, but, sincerely, we cannot adopt an experimental standard and/or a clinical purpose in a research with different a perspective.
Best Regards
The Authors